# *In Silico* Strategies in Tuberculosis Drug Discovery

**DOI:** 10.3390/molecules25030665

**Published:** 2020-02-04

**Authors:** Stephani Joy Y. Macalino, Junie B. Billones, Voltaire G. Organo, Maria Constancia O. Carrillo

**Affiliations:** 1Chemistry Department, De La Salle University, 2401 Taft Avenue, Manila 0992, Philippines; stephanimacalino@gmail.com; 2OVPAA-EIDR Program, “Computer-Aided Discovery of Compounds for the Treatment of Tuberculosis in the Philippines”, Department of Physical Sciences and Mathematics, College of Arts and Sciences, University of the Philippines Manila, Manila 1000, Philippines; vgorgano@up.edu.ph (V.G.O.); mtobrerocarrillo@up.edu.ph (M.C.O.C.)

**Keywords:** tuberculosis, druggability, docking, pharmacophore, MD simulation, QSAR, DFT

## Abstract

Tuberculosis (TB) remains a serious threat to global public health, responsible for an estimated 1.5 million mortalities in 2018. While there are available therapeutics for this infection, slow-acting drugs, poor patient compliance, drug toxicity, and drug resistance require the discovery of novel TB drugs. Discovering new and more potent antibiotics that target novel TB protein targets is an attractive strategy towards controlling the global TB epidemic. *In silico* strategies can be applied at multiple stages of the drug discovery paradigm to expedite the identification of novel anti-TB therapeutics. In this paper, we discuss the current TB treatment, emergence of drug resistance, and the effective application of computational tools to the different stages of TB drug discovery when combined with traditional biochemical methods. We will also highlight the strengths and points of improvement in *in silico* TB drug discovery research, as well as possible future perspectives in this field.

## 1. Introduction

In 1882, *Mycobacterium tuberculosis* (*Mtb*) was identified by Robert Koch as the causative agent of tuberculosis (TB), an infectious disease that continuous to be a relevant threat to global public health, especially in low- to middle-income countries. The pathogenesis of TB has several risk factors, including HIV infection, malnutrition, air pollution, type 2 diabetes, alcoholism, and smoking [1,2,3,4,5]. 

TB is encountered either as latent TB infection (LTBI), which is non-communicable and asymptomatic [6], or active TB, which is communicable and has symptoms such as fever, weight loss, productive cough, and hemoptysis [7]. Active infection is also classified depending on the strain: (1) drug-sensitive, (2) multidrug-resistant TB (MDR-TB), which is resistant to isoniazid and rifampicin, and (3) extensively drug-resistant TB (XDR-TB), which shows resistance to isoniazid, rifampicin, any fluoroquinolone, and aminoglycoside. Around 1.7 billion people are projected to suffer from LTBI and are at risk of progressing into active TB infection [8]. The World Health Organization (WHO) stated that active TB disease can be found in approximately 10 million people and has caused approximately 1.5 million deaths in 2018. An estimated half million individuals have rifampicin-resistant TB (RR-TB), of which 78% had MDR-TB. Furthermore, approximately 6.2% are suggested to have XDR-TB from these MDR cases [8].

## 2. Current Tuberculosis Management

One of the major challenges in managing TB is the estimated three million ‘missing’ individuals who have developed active infections but remained undetected or undiagnosed. TB can be deadly if not treated. With the help of conventional regimen, an estimated 58 million infected individuals were saved from 2000 to 2018. Global treatment outcome in 2017 shows a success rate of 85% for new TB cases and 56% for those with drug-resistant TB [8].

### 2.1. Latent Tuberculosis Infection

Treatment for LTBI are only provided for select groups that have a high risk of transitioning to active TB infection, including HIV-positive patients, people who were exposed to those with active TB, patients undergoing dialysis for end-stage renal disease, taking anti-tumor necrosis factor (TNF) medications, preparing for transplant surgery, or those with silicosis. Depending on whether it is beneficial or not, especially for children below 5 years of age, exposure to patients with active MDR-TB would require personalized treatment regimens and close observation. WHO recommended several different treatment regimens for LTBI, including 3 months of rifapentine and isoniazid, 3–4 months of isoniazid and rifampicin, 3–4 months of rifampicin, and 6–9 months of isoniazid [9,10]. While all these have established efficacy, poor patient compliance continues to be an issue especially with long treatment periods [9,10,11]. 

### 2.2. Active Drug-Sensitive Tuberculosis

In the last several decades, the treatment strategy for active drug-sensitive TB has not changed from the standard regimen of first-line drugs rifampicin, isoniazid, pyrazinamide and ethambutol (Figure 1) for the first 2 months continued by isoniazid and rifampicin for the next 4 months [12,13]. While this treatment procedure is highly efficacious and successful, its long duration primarily leads to poor patient compliance. This has long been an issue in TB management, necessitating monitoring protocols like the directly observed therapy (DOT), wherein a health professional directly supervises each dose intake [14]. Another issue brought about by the prolonged treatment is drug toxicity resulting in numerous adverse effects such as skin rash, gastrointestinal intolerance, neuropathy, arthralgia, increase in liver enzymes, hepatitis, immune thrombocytopaenia, agranulocytosis, haemolysis, renal failure, optic neuritis, and ototoxicity [15,16]. 

### 2.3. Multiple and Extensively Drug-Resistant Tuberculosis

Failure to complete the full TB regimen leads to disease relapse and drug resistance, which is more challenging to treat. A specific regimen can be designed depending on the resistance profile of the TB strain in a patient [17,18]. These treatments are often of longer duration (18 months or more) and utilize the more expensive second-line drugs (Figure 1) which have uncertain efficacy and high toxicity, resulting in poorer compliance and undesirable outcomes. To mitigate these issues, an updated seven-drug regimen guideline for the treatment of drug-resistant TB lasting 9 to 12 months was released by the WHO last 2016 [19]. 

With the increasing threat of treatment-resistant TB infection, a number of drugs have been fast-tracked to aid with the efforts in controlling TB worldwide. At the end of 2012, the US Food and Drug Administration (FDA) conferred accelerated approval to the drug bedaquiline for the treatment of resistant TB [20]. Bedaquiline’s anti-mycobacterial activity is due to its inhibition of the mycobacterial ATP synthase, a key enzyme in ATP synthesis, resulting in bacterial death. However, its use was shown to have an increased risk of death, thereby causing concerns about its approval. During clinical trials, roughly 11.4% of patients who took bedaquiline died as compared with 2.5% of those who took placebo treatments [21]. In 2014, the use of delamanid, a nitro-dihydro-imidazooxazole derivative, in the treatment of MDR-TB in adults was given conditional approval by the European Medicines Agency (EMA) [22]. Delamanid inhibits mycolic acid biosynthesis to block the formation of mycobacterial cell wall leading to improved drug permeation and more effective treatment [23]. Just recently, pretomanid in combination with bedaquiline and linezolid has also been approved by the FDA for treatment-resistant TB [24]. Pretomanid is a prodrug activated by nitroreductase, which reduces pretomanid’s imidazole ring to generate active metabolites. Specifically, a des-nitro metabolite leads to elevated levels of nitric oxide, which displays antimycobacterial activities due to its work as a poison for bacterial respiration under anaerobic conditions [25]. In aerobic conditions, it works like delaminid by targeting cell wall mycolic acid biosynthesis [26], and while there were several potential targets for this drug, its exact protein target is not yet known [27].

An increasing number of XDR-TB cases, such as in India, China, South Africa, Russia, and in eastern Europe, have proved difficult to treat even with the more intensive drug-resistant TB treatment regimen [18]. Novel therapeutics such as bedaquiline, delamanid, and pretomanid might help in curing these patients, though a suitable treatment regimen still has to be carefully designed. However, there is an additional difficulty in acquiring these drugs, especially in developing countries, resulting in a pool of patients that may remain untreated. Essentially, TB can be cured completely with the use of currently available and newly approved anti-tubercular drugs. However, difficulties in diagnosing and reporting infection, long treatment durations leading to drug toxicity and poor patient compliance, emergence of drug resistant strains, and limited acquisition of required treatment urgently necessitates the discovery and development of newer and effective drugs for TB.

## 3. Rise of Computer-Aided Drug Design in TB Drug Discovery

The drug discovery paradigm covers a wide range of fields, including biochemistry, chemical and structural biology, chem- and bioinformatics, computational chemistry, physical chemistry, organic synthesis, and others. The whole process entails large investments of time, money, and effort in order to produce promising candidates for the pipeline. Over the years, the drug discovery process for new antitubercular therapeutics have changed due to the increase in biological and chemical data, number of identified and validated targets, and advances in high-throughput screening technologies and software development. Moreover, the progress in data storage capacities, supercomputing powers, and parallel processing in the last several years allowed computer-aided drug design (CADD) to become an integral part of TB pharmaceutical research. This continuing expansion in computing power can soon potentially allow the exploration of the vast chemical space, thought to comprise of approximately 10^60^ organic molecules below 500 Da, in order to identify therapeutically interesting scaffolds [28]. Moreover, the boom in protein structural data, including over 150,000 macromolecular structures found in the Protein Data Bank (PDB, www.rcsb.org) [29], proved beneficial in elucidating important molecular and computational concepts for drug design studies. As with any other disease, TB has been the subject of continuous and numerous drug discovery studies, including thousands of published CADD investigations. Despite this, a paper by Ekins et al. noted gaps in the application of these methods in TB research [30], resulting in the slow output of candidates into the TB drug pipeline despite the apparent need and urgency for this disease. This suggests that more rigorous efforts are needed in TB drug discovery to maximize the advantages provided by computational tools.

Computational or *in silico* methods are knowledge-driven, rationally exploring available data to investigate protein function and design new molecular entities (NMEs) that can effectively regulate its behavior. Computational drug discovery approaches are generally divided into structure—(SBDD) and ligand-based drug design (LBDD), depending on the availability of structural data (Figure 2). However, it has been a common practice to integrate these methods in a complementary manner in order to increase the success rate of current drug discovery projects (Figure 2). SBDD requires the target’s three-dimensional (3D) structure to be able to examine and use the binding pocket for screening and design of suitable ligands, which can then be experimentally validated and optimized. In the absence of protein structural data, LBDD utilizes knowledge gained from a collection of diverse ligands with known activity to create predictive models for hit discovery and lead optimization [31]. Different types of SB and LB strategies, or a combination thereof, can be applied at different stages of TB drug discovery and development in order to alleviate the challenges involved with experimental methods. With the availability of TB genome and proteome, as well as abundant structural data, data mining and docking strategies can be employed for target identification. Virtual screening (VS) can then be applied to pick out the best potential candidates from a database containing millions of molecules for a chosen TB target. After validation of candidates, structure-activity or -property relationship (SAR/SPR) studies can be implemented to understand mechanism of action and ADMET (absorption, distribution, metabolism, excretion, and toxicity) properties in order to design compounds with better activity and pharmacokinetics. Data (both positive and negative results) taken from these investigations can be kept and used for further iteration and method optimization in the design of novel TB compounds. Both commercial and free software and webservers have been developed covering different SBDD and LBDD techniques, some of which are listed in Table 1.

### 3.1. Databases

The era of big data has greatly affected the current drug discovery paradigm through innovations in data storage, management, and mining. Moreover, drastic cost reductions in sequencing technologies allowed the study of multi-omics (e.g., genomics, transcriptomics, proteomics, and metabolomics) for several species including *M. tuberculosis* [124,125,126,127,128,129]. In order to take advantage of the benefits provided by SB and LB techniques, these biological and/or chemical data must be acquired for analysis via numerous publicly accessible databases on the internet [130,131,132]. Given that TB is an old disease, vast amounts of data points have already been gathered and are waiting to be used in the fight against this infection. 

One of the most extensive and widely-used protein information resource is UniProt (https://www.uniprot.org/) [133], which consists of annotations from several other databases for protein function, omics, and structural data. More specific to TB, the TB Database (http://tbdb.bu.edu/tbdb_sysbio/MultiHome.html) [134,135] contains information on mycobacterium genomes, genes, gene expression correlation, gene epitopes, and experimental and computational models of TB molecular pathways. Alternatively, genomic and proteomic data for various pathogenic mycobacteria can also be found in the Mycobrowser (https://mycobrowser.epfl.ch/) [136], which is linked to UniProt for mycobacterium protein information. On the other hand, patient clinical data is provided by the TB Portals (https://tbportals.niaid.nih.gov/) [137], which is an open-access platform containing socioeconomic, geographic, clinical, laboratory, radiological, and genomic data from patients infected with drug-resistant TB, from the National Institute of Allergy and Infectious Diseases (NIAID) in collaboration with data scientists and clinicians and scientists from countries suffering from heavy TB burden.

Advances in structural and computational biology techniques led to the surge in structural data, resulting in thousands of three-dimensional protein structures generated from X-ray crystallography, nuclear magnetic resonance (NMR), cryo-electron microscopy (EM), homology modeling, and molecular dynamics (MD) simulations. Data from these experiments are customarily deposited to structure databases such as PDB [29], PDBsum [138], etc. Associated with this, the size of the virtual chemical space [28] and improvements in combinatorial chemistry [139] also permitted the availability of chemical libraries (Table 2). 

Protein subcellular localization databases are also available for study, such as eSLDB (eukaryotic Subcellular Localization database) for general eukaryotes [146], LOCATE for human and other mammals [147], and PSORTdb for bacteria and archaea [148]. Lead optimization and drug repurposing researches can also benefit from protein binding databases like ReLiBase, which consists of interaction information for receptor-ligand complexes from PDB [149], BindingDB, which describes interactions and affinity information between protein target and drug-like molecules [150,151], and Database of Interacting Proteins (DIP) [152], Biological General Repository for Interaction Datasets (BioGRID) [153] and Search Tool for the Retrieval of Interacting Genes/Proteins (STRING) [154], which contains data on protein-protein interactions. 

### 3.2. Structure-Based Tools

#### 3.2.1. Comparative Modeling, Binding Site Prediction, and Druggability 

Employment of SBDD tools (Figure 2 and Table 1) require not only the availability of 3D structural data but also information on its druggability and potential binding sites. In the absence of structural data obtained from experiments, such as X-ray crystallography and NMR, a computational model can also be generated either through homology modeling or protein threading techniques (Table 1), which are well-established methods in protein comparative modeling. Homology modeling entails the use of a structural template with suitably similar sequence as the target protein [155]. The most critical stage of any homology modeling procedure is the initial sequence alignment. While there are numerous bioinformatics tools available for this, such as NCBI Blast [156,157,158], COBALT [159], Clustal Omega [160], KAlign [161], etc., manual inspection and modification of the alignment is crucial, especially if a researcher’s knowledge about specific protein folds and domains need to be further incorporated. Then, the secondary structures (i.e., alpha helices, beta strands, loops, etc.) are copied from the template based on the final sequence alignment in order to approximate the target structure. The final model is then refined through minimization or MD and its stereochemical quality is checked using tools like those listed in Table 1 until the structure has improved and is suitable for further computational studies. 

Druggability is the capacity of a protein target to be modulated by a ligand. It is important to characterize this property as it helps avoid intractable proteins and allows the identification and prioritization of significant targets. Some predictive approaches that include this property are listed in Table 1. A druggability database, the Druggable Cavity Directory (DCD), is also publicly available to allow researchers to submit protein pocket and druggability information, which is later verified and made available to other researchers [162]. After ascertaining that a given target is indeed tractable, binding pocket information should be acquired either from protein structures complexed with natural substrates or known inhibitors, or from mutational data distinguishing key interaction residues. Ideally, a binding pocket is a concave area in the receptor that is characterized by chemical features with which a ligand can desirably interact to attain the required receptor behavior (e.g., inhibition or activation) [163]. However, if binding information is unknown, several *in silico* methods and webservers (Table 1) are available to identify potential receptor binding sites from a given structure and have been described in detail elsewhere [31]. Otherwise, a number of studies have also used ‘blind’ docking [164], wherein the whole protein is set as the binding site, allowing ligands to freely bind anywhere in the structure in the hopes of finding a suitable pocket. 

It is also prudent to remember that other potential binding sites may be present on the target surface, i.e., allosteric sites. Conventional drug discovery efforts often target the primary (orthosteric) binding site to block substrate binding. But, as in the case of uncompetitive and noncompetitive inhibitors, ligands can also allosterically modulate activity within the protein structure. Such is the case in the study done by Shi and colleagues, where they identified a second druggable binding site in *Mtb* UDP-galactopyranose mutase (UGM) [165]. NMR and kinetics studies classified MS-208, a known *Mtb*UGM inhibitor, as a noncompetitive/mixed inhibitor and therefore binds in another site in the enzyme to allosterically affect substrate binding. Blind docking in AutoDock Vina [63] was performed to identify possible allosteric sites for MS-208. Two regions, A- and S-site, were initially identified and docked complexes featuring binding to either sites were further subjected to simulation studies using Amber [81]. The A-site-bound structure exhibited the most stable complex formation with excellent interaction energy, as well as the most number of contacts, suggesting that this is the second druggable binding site in *Mtb*UGM [165].

#### 3.2.2. Pharmacophore Modeling and Molecular Docking

Virtual screening is one of the most popular *in silico* drug discovery approaches as it allows researchers to quickly extract data from unexplored chemical space in a cost-effective manner. It has become customary to complement high-throughput screening with VS for the prioritization and identification of novel ligands with the most potential as starting points for drug discovery efforts [166]. Two of the most common VS methods are pharmacophore modeling [167] and docking [168]. 

Pharmacophore models can be generated from a receptor alone or a receptor-ligand complex. Previously, due to lack of protein structural data, ligand-based pharmacophore has been more customarily used (see Section 3.3.2). A pharmacophore is a group of geometrically-mapped chemical features, such as *H*-bond donors and acceptors, hydrophobicity, and ionizable groups, required for optimal interactions to elicit a response between a receptor target and its partner molecule [169,170]. In line with 3D-mapping in a pharmacophore, exclusion volumes can also be included in the model to incorporate binding site shape [171]. After generating a 3D model, large chemical databases can be efficiently searched for candidates that match pharmacophore elements. However, the pharmacophore database screen only provides a fit score that cannot be translated as affinity. The fit score weighs the alignment quality between the ligand substructures and center of model features. Weights and penalties can also be employed for features deemed significant to activity [171]. These models can also be used for scaffold hopping, allowing for the discovery of novel chemotypes based on fit of interaction and geometric characteristics [171,172].

Another well-established *in silico* method is docking, which can be used to facilitate the investigation of how ligands can fit and complement receptor binding pocket features in order to modulate its activity [168]. Numerous docking methods (Table 1) have been developed and comparative studies and detailed reviews about these have been published elsewhere [173,174,175,176,177]. In the early days of computational drug discovery, docking was developed to be able to predict the bioactive conformation within a set of docking results. However, protein-ligand interactions need to be evaluated using a scoring function to find the best pose using estimated affinity, distinguish actives from inactives, and prioritize candidates for further testing and optimization. This was soon discovered to be the most challenging part of docking due to approximations applied to other crucial factors, such as protein flexibility, solvent involvement, and system entropy [168]. Despite advances in scoring functions through increased understanding of protein-ligand interactions, it is difficult to handle all these aspects while still maintaining method efficiency. Moreover, scoring functions often depend on the protein families and ligand sets from which it was generated and validated [31,178]. And while there is currently no ideal scoring function that can be utilized across all druggable targets, implementation of method validation before starting any VS project establishes whether a chosen docking method and scoring function can be applicable or not [171]. 

Both pharmacophore modeling and docking have been applied in combination with other *in silico* tools for the identification of novel antimycobacterial agents. Pharmacophore screening followed by docking can be employed as complementary screening tools, resulting in faster processing and more optimized results. A recent study published by our group exemplifies both pharmacophore-based and docking VS by targeting *Mtb* 7,8-diaminopelargonic acid aminotransferase (BioA), an important enzyme in its lipid biosynthesis pathway with no corresponding human ortholog [179]. A receptor-based pharmacophore was generated in Discovery Studio [180] using the BioA structure, characterizing 25 functionalities (nine hydrophobic, nine H-bond donors, and seven *H*-bond acceptors), and was employed to screen 4.5 million compounds from the Enamine REAL database. Compounds with good pharmacophore fit, as well as the co-crystallized inhibitor, were subsequently docked to the BioA protein via CDOCKER [70] and ligands with better binding energy values than the known inhibitor were chosen for the TOPKAT protocol [109] to filter out potentially toxic compounds. This step-by-step screening led to the identification of 45 virtual hits, 17 of which were available for purchasing and validation. Whole-cell assay was performed to eliminate compounds that cannot penetrate the distinctive thick, waxy lipid layer of the mycobacterium, identifying compound **7** ((*Z*)-*N*-(2-isopropoxyphenyl)-2-oxo-2-((3-(trifluoromethyl)-cyclohexyl)amino)acetimidic acid) as a potential BioA inhibitor with a minimum inhibitory concentration of ~25 µM [179]. 

Throughout the years, various improvements have been harnessed to enhance ranking performance, such as rescoring or consensus scoring. Given the different strengths and limitations of each scoring function, rescoring with the help of a separate scoring function not used in a docking study provides users with a different perspective for selection of final hits. For instance, a faster scoring function can be employed for pose prediction while another one is used for affinity prediction and ranking [171]. Currently, consensus scoring is more commonly used for docking studies and has already been examined by several groups in the last couple of decades as an improved protocol for finding potential hits [181,182,183,184,185]. This strategy aims to characterize the intricacy of target molecular recognition based on various energy functions which can be covered by several scoring schemes, resulting in decrease in false positives [184]. However, there is also a risk of rejecting true positives, which have favorable scores in only 1 function used. Thus, it is also imperative to validate a consensus scoring workflow against specific targets [171]. An exemplary case features salicylate synthase MbtI, a critical enzyme in the biosynthesis of siderophore mycobactins, which is used by *Mtb* to chelate iron required for growth and survival in the host. Absence of siderophores prevents bacterial growth in the persistent state after engulfment by macrophages [186]. Previously identified MbtI inhibitors include those based on the MbtI reaction intermediate isochorismate [187], benzimidazole-2-thione [188], and chromane scaffolds [189]. Chiarelli et al. [190] discovered furan-based MbtI inhibitors through structure-based pharmacophore and consensus docking. The pharmacophore model was generated from important binding features in the MbtI-inhibitor complex, including interaction with the conserved Y385, lipophilic interactions, and ionizable interaction with the Mg^2+^ ion. Screening of 1.5 M compounds from Enamine [141] led to over 2,000 pharmacophore hits which were subjected to consensus docking. Docking methods including AutoDock [62], AutoDock Vina [63], DOCK [64], FRED [191], GOLD [65] (comprising four scoring functions), and PLANT [192] were first evaluated using several MbtI-inhibitor complexes to identify which methods are most reliable. GOLD and PLANT were employed for the consensus procedure of the pharmacophore hits, and those with similar binding modes and consistent scores across all scoring functions were further examined if the docked conformation still matched the 3D arrangement of the pharmacophore model. From these, five virtual hits progressed to bioassays, wherein two compounds showed potent MbtI inhibitory activity. MD simulation was additionally applied to study enzyme-ligand interactions and provide information for further optimization. The furan scaffold from the more potent hit was used as a starting point for lead optimization, resulting in a candidate with promising activity against MbtI and suitable antimycobacterial activity [190].

Inclusion of limited protein flexibility, such as in the binding pocket, while still maintaining efficiency has been considered in methods like induced fit docking (IFD) and ensemble docking. IFD incorporates the principle that ligand binding induces changes in residue side chain conformations within the specified pocket, thereby inciting tighter binding with the receptor [193,194]. However, backbone movement should also be considered as it increases the accuracy of side chain positioning and orientations [31]. An example of this is shown in Figure 3, in which *Mtb* InhA exhibits backbone and side chain conformational differences between its apo (PDB ID: 4DRE) [195], fatty acyl substrate-bound (PDB ID: 1BVR) [196] and isoniazid (INH)-bound (PDB ID: 4TRO) structures [197]. These movements can change the binding pocket shape and volume, potentially affecting VS results. The utilization of several experimentally—(X-ray crystallography, NMR, or cryo-electron microscopy) or computationally-derived (MD trajectory) protein structures to integrate both backbone and side chain movements has also been employed in ensemble docking [198]. Structural ensembles provide better reproducibility of experimental conditions as rigid protein structures, such as those obtained from X-ray crystallography experiments, only provide a snapshot of a dynamic ensemble of conformations.

Various docking approaches were applied by Brindha et al. [199] for drug repurposing against *Mtb* murE, which is an attractive target due to its significance in the peptidoglycan biosynthesis of tuberculosis bacteria and lack of eukaryotic homolog. VS of compounds from DrugBank [200] was first implemented through the parallel use of Glide Standard Precision (SP) [66] and AutoDock Vina [63]. To improve prioritization of compounds through the incorporation of binding site flexibility, common hits from both methods were further subjected to IFD [67]. Final rankings were done using Glide eXtra Precision (XP) scoring and AutoDock Vina binding energy prediction, resulting in 17 common top hits identified as repurposed antitubercular drug candidates [199]. In another example, ensemble docking using three enzyme structures was performed to better elucidate ligand binding interactions, especially due to the binding site flexibility of the *Mtb* Type II dehydroquinase (MtDHQase), an essential virulence factor in TB [201]. Conformation of key residues were determined by analyzing superimposed MtDHQase structures and rotamer distribution of each residue from the penultimate rotamer library [202]. The benzene sulfonamide containing compound with the best activity, a Schaeffer’s acid amide, was docked using GOLD [65] and scored using ChemPLP [203]. Varying the side chain flexibility during the docking procedure led to the identification of residues that are required to transition the binding site into an open conformation, which is the preferred conformation of the inhibitor to display its activity. Along with the interaction and flexibility data, Schaeffer’s acid amide can be optimized into a potent antitubercular therapeutic compound [201].

#### 3.2.3. Molecular Dynamics

The availability of 3D structural information has greatly helped in structure-based drug design by presenting atomic-level insights into molecular interactions. Nonetheless, these provide only partial interpretations of biomolecular structures, as well as related aspects of molecular recognition and binding. In physiological conditions, proteins frequently undergo conformational changes upon binding with a partner, such as a small molecule, peptide, or another protein, to perform a specific function. At times, these transformations only involve side chain conformations and small to medium movements in the backbone. However, there are cases in which significant deviations are seen in the overall protein fold and/or subunit arrangement [204,205,206,207]. 

Molecular dynamics simulation, a method that was first developed in the 70s [208], can be employed to analyze these protein dynamics and study the binding energy landscape. The availability of MD platforms (Table 1) allowed for the routine assimilation of simulation studies for systems containing ~50,000–100,000 atoms. The investigation of even larger systems is made possible using graphics processing units (GPUs), which are high-performance processors that can support heavy computational load, and high-performance computing (HPC) technologies featuring messaging passing interface (MPI), a system which employs multiple cores in parallel to distribute computational load and reduce the time required for simulation [206,209]. Popular MD packages have been adapted for these tools, and while MD simulation projects commonly use a combination of both, the speedy development of more advanced GPUs increasingly allows for the use of personal workstations [206].

To start an MD simulation, a 3D protein structure of the required system (e.g., apo protein, protein-ligand or protein-protein complex) must be obtained experimentally or through homology modeling, and represented based on the duration and details of study [206,210]. Another critical aspect of system preparation in MD is the solvent model, which can be explicit or implicit. Explicit solvent is more frequently used due to its simplicity and its proficiency in recovering native solvent effects to protein structure [211]. However, large system size resulting from this model makes conformational sampling challenging. To speed up conformational sampling, an implicit solvent model can be generated by adding approximations to the system, but this may affect the free energy landscapes [212]. Once a solvent model has been chosen, the next step is to select an appropriate force field, which is used to define the forces acting on every atom in the system and to calculate the potential energy within the molecular structure. While there are numerous force fields that have been and are still being developed and improved, some of the most popular force fields applied in simulation systems are currently CHARMM [213], Amber [214], GROMOS [215], and OPLS-AA [216,217]. Different force fields use different parameterization to characterize atomistic molecular simulations, distinguishing their applicability in atomistic molecular simulations of diverse target structures and systems, but are often equivalent [206]. To ensure efficiency while keeping the calculations accurate, simple molecular representations in force fields include springs depicting bond length and angles, periodic functions depicting rotations and Lennard-Jones potentials, and Coulomb’s law characterizing van der Waals and electrostatic interactions within the system [206]. Newton’s law of motion is then employed for the computation of accelerations and velocities during atom movement. Minimization and equilibration are typically performed ahead of a production run to adjust the system to the applied force field, relax steric clashes, and to stabilize system temperature and pressure. Once the prepared system is correctly minimized and equilibrated, the production run can be performed for a suitable amount of time (ps, ns, μs, etc.) depending on research needs (Figure 4). A timestep of 1 or 2 fs is frequently used for atomistic MD simulations [206].

After obtaining simulation trajectories, this information can be used for additional analyses, including but not limited to: (1) verifying stability through root-mean-square deviation (RMSD), root-mean-square fluctuation (RMSF), or radius of gyration (RoG) [218,219], which can identify critical components for protein flexibility, (2) investigating protein structural or energy networks through network analysis [220], which can pinpoint residues that are pivotal in allosteric or long-range communications within the protein, (3) studying protein energy landscape by mapping trajectories [221], providing information about protein folding and function, as well as most stable populations within a given trajectory. Findings from each analysis provide crucial information that would otherwise be imperceptible with other techniques, thereby increasing our understanding of a given system.

With current computational capabilities, simulation times are nearly of biological relevance, allowing researchers to observe biological events, such as allosteric regulation, transient protein changes and binding, and enzyme catalysis. A number of MD studies have already been done for the structural and functional elucidation of validated TB targets and design new antitubercular agents [222,223,224]. One such study investigated the differences of inhibitor binding against wild-type and mutant structures of InhA, a very well-known TB target [223]. Mutations for this protein led to lower affinity for its co-factor, NADH, resulting in isoniazid resistance. MD simulations of the wild-type and mutant structures of InhA bound to NADH were performed using Amber [81] to understand the underlying aspects affected by structural mutations. Schroeder and colleagues found that mutations in the glycine-rich loop (I21V and I16T) disturbed the NADH binding conformation, specifically that of its pyrophosphate moiety, and decreased its direct and indirect *H*-bond contacts within the binding pocket. Isoniazid requires the formation of a covalent adduct with NADH within the InhA binding site. Changes in binding interactions and conformation of NADH can negatively affect this, hence, contributing to isoniazid resistance [223]. A very recent MD study for TB drug discovery using Amber [81] has been published by Cruz et al. [224], wherein the binding mechanisms of Tam1 and its analogs against polyketide synthase 13 (Pks13), an enzyme that carries out the final step in mycolic acid biosynthesis [225], were investigated to obtain insights that can aid in the design of new antitubercular agents [224]. Fluctuation analysis revealed distinct flexibility in the protein lid domain of *Mtb* Pks13, which was decreased upon ligand binding, suggesting that residues from this domain are critical for ligand interaction. Binding free energy calculations from trajectory data agreed with experimental data, identifying Tam16 as the most potent of the Tam analogs due to conformational stability offered by *H*-bond interactions at both ends of the ligand structure that was not observed for other compounds. Energy decomposition analysis further specifically identified residues that greatly contributed to inhibitor binding, which can then be targeted for optimization of Tam16 and the design of other analogs [224]. 

### 3.3. Ligand-Based Tools

#### 3.3.1. Similarity-Based and Quantitative Structure-Activity/Property Relationship Methods

Even before the upsurge of available target structural data, rational inhibitor design has been employed with the help of substrate or product structures, and thus termed as ligand-based drug discovery and design (Figure 2). The simplest and most inexpensive LBDD approach is the similarity-based method, wherein compound candidates were designed and optimized through the principle of chemical similarity, i.e., similar (untested) ligand structures are posited to have similar activities as known inhibitors or modulators [226,227]. A typical workflow requires one or more reference structures with existing bioactivity information against a specific target. This is then used as a template to select new potential candidates from a chemical database to prioritize for assay testing. 

Different molecular descriptors and parameterizations can be employed to characterize compounds and efficiently determine similarities. Descriptors can be generated as one-, two- or three-dimensional (1D, 2D, or 3D), wherein 1D descriptors comprise of global ligand properties (e.g., molecular weight, logP, number of *H*-bonds, etc.), 2D descriptors include topological and connectivity properties (e.g., aromaticity, degree of branching, etc.), and 3D descriptors involve geometrical properties (e.g., shape, volume, surface area, etc.) [228,229]. Additionally, fingerprints can also be used to depict template and database compounds by rendering structural features, such as those based on substructure (i.e., scaffold or functional groups), topology or path (i.e., fragmentation following a linear path of bonds), circular or radial (i.e., surrounding features of an atom up to a certain radius), and pharmacophoric (i.e., distance-based features, incorporating molecular shape, and interactions required for biological activity) elements [230]. Well-known platforms that generate molecular descriptors and fingerprints are listed in Table 1.

To compare structural similarity after obtaining simplified molecular features, similarity coefficient and weighing scheme are required to measure and highlight the importance of certain aspects of a compound’s structure in relation with its activity. Given that there are multiple tools available for both components, careful selection of analysis tools is crucial to have successful VS campaign. Tanimoto coefficient, which uses the ratio of shared features in both fingerprints to the total number of features between each fingerprint sets, is used as a standard for similarity evaluation of any two vectors. This coefficient returns values between 1 and 0 to depict chemical similarity [231]. Other known similarity measures include Manhattan distance, Euclidean distance, Dice index, Cosine coefficient, and others [232]. In terms of weighing schemes, some features may be ‘silenced’ or set as optional depending on its importance for a specific activity. There is not one method that can be considered the best for the full range of known targets and chemicals, as each method have their own data set applicability [233,234]. In this case, data fusion can be employed to obtain a consensus of outputs from different methods [235,236]. Both 2D and 3D similarity methods have been successful in identifying hits for various targets and have been established to have comparable or even better enrichment than docking [237]. While similarity-based approaches are known for their efficiency, there is a risk of obtaining low diversity hits as most similarity methods are highly dependent on the input structures used to calculate descriptors [238]. Moreover, there is a potential occurrence of ‘activity cliffs,’ in which small modifications in a ligand structure lead to significant difference in activity [239,240,241]. 

The similarity concept is also applied in studies involving quantitative structure-activity/property relationship (QSAR/QSPR), a method which is used to investigate the correlation between structural and physicochemical properties of ligands with known biological activities. QSAR modeling depends on the premise that ligand 2D and 3D properties can provide information to establish a statistical model of the desired biological activity, which can then be employed for activity prediction of ligand candidates [242,243]. The statistical model is generated based on an appropriate data set, consisting of compound structures with known bioactivity against a specific target, which must be checked and pre-processed. This data set should contain an adequate number of samples (i.e., a minimum of 20 experimentally-validated compounds) and, if from separate studies, identified using same assay protocols such that equivalent activities are obtained. Included in the data set preparation, especially for higher dimensions of QSAR modeling, is the conformational selection and alignment which allows the identification of scaffolds and functional groups that are critical to activity and therefore has more weight in the statistical model [242]. In this case, it is important to remember that the lowest energy conformation is not always equivalent to the bioactive conformation [244,245] and that ligands chosen for the training set should interact with the same binding site [242]. Typical alignment methods include the analysis of molecular fields, structural shapes, or pharmacophores. Pharmacophore generation for ligand alignment is more favorable as it aligns compounds based on feature similarity rather than chemical substructure [246]. As with similarity-based methods, QSAR makes use of molecular descriptors with dimensionality depending on the information available. Descriptors applied to the model should be carefully chosen to avoid autocorrelation and over-fitting. Before proceeding with and *in silico* prediction study, the model must be validated with internal and/or external data sets to establish its predictivity and applicability against a desired target [31,247]. 

Different QSAR methods have been developed and incorporated in various open-source platforms and commercial software (Table 1). The earliest QSAR-based algorithms include Comparative Molecular Field Analyses (CoMFA) [248] and Comparative Molecular Similarity Indices Analysis (CoMSIA) [249], both of which are still widely used today for various drug discovery endeavors [250,251,252,253]. 3D-QSAR CoMFA was used by Singh and Supuran [252] for the discovery of novel *Mtb* carbonic anhydrase inhibitors. A number of sulfonamides that target *Mtb* carbonic anhydrase 2 to regulate bacterial growth were used as the data set for QSAR modeling. The best developed model had excellent predictivity and good fit with an r^2^ value of 0.93 and cross-validated coefficient q^2^ value of 0.88. From the CoMFA results, it was also determined that several steric and electrostatic features play critical roles in the inhibition of *Mtb* carbonic anhydrase 2. Using this information, nine compounds were designed and later observed to have better predicted inhibitory activities compared to the test set used. However, experimental validation is still required to determine the feasibility of these findings [252].

#### 3.3.2. Ligand-Based Pharmacophore Modeling

Pharmacophore modelling have already demonstrated its value in ligand-based drug discovery studies. Ligand substructures required for optimal bioactivity can be aligned and characterized as a spatial arrangement of features in 3D space, which can be directly used for screening or applied to 3D-QSAR modeling [169,170,246]. Several software and webservers (Table 1) are available for the generation of ligand-based pharmacophores. 

As with any other ligand-based methods, a data set of diverse ligands with known bioactivity against a specific target is required for pharmacophore generation. The training ligands used for ligand-based pharmacophore modeling must bind to the same pocket and have similar binding interactions, much like in QSAR studies. After obtaining a pharmacophore, its validity is assessed using a separate test set. It is then employed for virtual search of candidate compounds from libraries of untested molecules, wherein compounds are taken as potential hits if it ‘fits’ well with the pharmacophore (Figure 5). The main advantage of pharmacophore modeling is the use of molecular features rather than structural groups in depicting critical functionalities for activity, which allows for the identification of novel ligands with diverse structures (i.e., scaffold hopping) [172]. Moreover, pharmacophores can also be used for target profiling and polypharmacological studies to avoid adverse effects resulting from off-target binding [254]. This is especially useful when designing antitubercular and other antibiotic or antiviral agents as to avoid harmful interactions with human proteins.

Due to the numerous parallels between pharmacophore and 3D-QSAR modeling, these methods have been used in combination for a number of ligand-based drug discovery efforts. A study by Tawari et al. used PHASE [80] to target the *Mtb* aryl acid adenylating enzymes known as MbtA, which are involved in siderophore biosynthesis in tuberculosis [255]. A set of nucleoside bisubstrate analogs with known whole cell assay activity and bioactivity against siderophore biosynthesis in *Mtb* were used for pharmacophore and QSAR model development. *H*-bond donor, *H*-bond acceptor, and aromatic features were found to be critical for the inhibition of MbtA. The pharmacophore was also used to align molecules for the 3D-QSAR model, which exhibited suitable predictability and applicability with a Q^2^ value of 0.71, RMSE of 0.65, and Pearson-R of 0.85 when assessed against a test set. The SAR studies additionally revealed the disadvantageous effects of bulky groups at the adenyl moiety *C*-6 position. Information taken from these models can be used for the rational design of new MbtA bisubstrate inhibitors as antitubercular agents [255].

#### 3.3.3. Density Functional Theory

Density functional theory (DFT) is quantum mechanical method established in the 1960s [256,257], which can be used in material science, computational chemistry, and computational physics to study the electronic properties of a many-particle (e.g., atom, molecule, condensed phase) system. DFT is based on two Hohenberg-Kohn (HK) theorems. First, the ground state properties of the many-particle system can be determined using only three spatially determined electron densities. Second, the HK theorem describes an energy functional, which can be minimized by the correct ground state electron density [258]. The use of DFT circumvents the computational expense of conventional methods like Hartree-Fock (HF) Theory since DFT relies on the premise that energies, intricate motions, and pair correlations can be derived directly from the electron probability density alone, instead of using wavefunctions. Theoretically, quantum mechanical wavefunctions consist of all the information required from a target system, and while the Schrödinger equation can be solved for a simple system, such as that of a hydrogen atom, it needs extensive computational efforts and it is impossible to solve this for a many-body system. In this case, DFT is used as an equivalent and efficient alternative to the Schrödinger equation [259], making DFT a popular tool in several computational fields [260]. 

In tuberculosis research, DFT has found uses in studies involving catalytic mechanisms [261,262], structure-activity relationship analysis [263], and inhibitor potency [264]. Chi and colleagues applied DFT to support their initial observations regarding a change in inhibitor binding mode in the MbtI enzyme after the addition of a substituted enolpyruvyl group to the parent compound structure previously designed from isochorismate [264]. X-ray crystal structures of MbtI complexed with its inhibitors depicted two different binding modes (Mode 1 and 2), suggesting binding site flexibility to accommodate ligand binding. The global minimum conformation of (*E*)-3-(1-carboxyprop-1-enyloxy)-2-hydroxybenzoic acid (AMT), *Z*-methyl-AMT, and *E*-methyl-AMT inhibitors in solution were calculated using Gaussian09 [265] with the B3LYP hybrid functional [266,267]. Global minimum conformation of free *Z*- and *E*-methyl-AMT were found to be similar to its bound conformation (Mode 2), indicating prearranged conformations to facilitate its binding to MbtI. Calculation of conformational entropy values for the three compounds revealed that *Z*-methyl-AMT is the least disordered, which may be due to the methyl conformational lock in its structure. Although a pure *Z*-isomer has not yet been obtained to experimentally differentiate it from the *E*-isomer, this finding rationalizes potent binding of methyl-AMT to MbtI and offers more information for the future design of novel and potent MbtI inhibitors [264].

Despite the success and popularity of DFT, it still has deficiencies due to approximations used in the development of functionals. Systems predominantly comprised of dispersion (van der Waals) forces, such as gaseous systems, or those wherein dispersion has a considerable contribution, such as biomolecular systems, are challenging to characterize using DFT [268]. However, several studies have already investigated the inclusion of van der Waals to improve this method [269,270,271]. Other major limitations of DFT application in computational chemistry include the characterization of charge transfer excitations, transition states, and global potential energy surfaces [272].

### 3.4. Integrated Tools

With the variety of available tools and structural data for drug discovery nowadays, it is more common to find studies that employ a combination of structure- and ligand-based approaches rather than exclusive application of each (Figure 2). Additionally, integration of these strategies often produces better results owing to more effective exploration of chemical and biological space. Moreover, the strengths of one method can overcome the limitations of the other, resulting in a highly complementary drug discovery process [171,238,273]. Integrated *in silico* workflows include sequential and parallel or data fusion methods [274,275,276], though hybrid methods have also already been developed [277,278]. Sequential methods involve the successive use of computational methods with the aim of increasing the selectivity of the VS workflow by continually reducing the number of potential hits before experimental evaluation [31]. However, it has been established that structure- and ligand-based methods have similar enrichment and frequently yield hits with different scaffolds [276], indicating that these methods are better applied in parallel rather than sequentially [279]. Parallel application, through simultaneous employment of various computational tools, often produce a more diverse hit profile [31,280]. Nonetheless, since results from these methods are often fused to produce a final ranking, a large number of virtual hits is obtained from this approach [31]. 

*In silico* methods have already been applied to tuberculosis studies in several different ways and combinations depending on the goal of the study implemented, such as for drug discovery [179,281], understanding protein structure and function [282,283], and others [284]. One example showing the integration of computational methods is a study implemented by Li et al. involving 3D-QSAR, binding pocket prediction, docking, and MD simulation studies for FtsZ, which is a validated *Mtb* target and plays a significant role in cell division [285]. Trisubstituted benzimidazoles were found to target this protein and used for 3D-QSAR CoMFA [248] analysis in order to elucidate important structural factors related to their inhibitory activities. Homology modeling using the SWISS-MODEL server [32] was required to obtain the GDP-bound structure of *Mtb* FtsZ using *S. aureus* FtsZ as template. Afterwards, binding site prediction using ProFunc [61] was performed to identify potential binding pockets (other than the GDP binding site) for the candidate compounds. Selected trisubstituted benzimidazole analogs were docked into the *Mtb* FtsZ model using AutoDock, after which the lowest binding energy docked complex was refined using MD simulation. Using the MD-refined *Mtb* FtsZ structure, all trisubstituted benzimidazoles were docked using Surflex-Dock [286]. In the lowest energy state of the compounds, the benzimidazole scaffold and cyclohexyl group were located in a highly hydrophobic pocket within FtsZ, while the carbamate groups were oriented towards the hydrophilic area. These interactions are posited to be crucial for ligand binding stabilization and inhibition of *Mtb* FtsZ. The results of this study display how the concerted application of different *in silico* methods can lead to better understanding of protein structure, ligand design, and inhibitory activities [285].

## 4. Edges and Pitfalls of *In Silico* Methods

There are roughly 2500 protein structures for tuberculosis in the PDB and perhaps thousands of ligand candidates published. All these pieces of information are available with a few keyboard strokes and a click of the mouse. Along with existing technologies, it is now possible to analyze TB enzymes and lead candidates at the atomic level in order to understand their function and how to regulate them. While computational methods have been widely used in drug discovery nowadays due to their successful applications [287,288,289], it is still important to remember that these tools are like any other experimental approaches—prone to limitations dependent on the system and other various parameters being studied [290,291,292]. 

VS has been known to successfully screen millions of compounds to identify potential inhibitors for a given target [287,289]. This lends efficiency to cost, time, and efforts used in drug discovery projects as only the most promising compounds are brought forward for more rigorous experimental testing and drug development. However, optimization and validation of these methods are far from perfect and are highly dependent on the protein system and compound classes used, leading to possible bias in the computational model. Thus, it is challenging to determine which method has the advantage over another; many benchmark studies have been published regarding this matter [293,294]. Other major limitations include difficulties in incorporating protein flexibility and solvent effects due to the computational burden attached to these factors [31]. Fortunately, available technologies seem to be catching up as enhance sampling methods, HPC, and MD platforms are now routinely applied in drug discovery projects and are known to calculate up to milliseconds of simulations for various protein targets [295,296,297]. In terms of ligand-based drug design, its main advantage is its simplicity and efficiency. Indeed, LBDD has a long history and numerous candidates have already been discovered even with the lack of protein structural information [298,299,300]. Nonetheless, several factors should be considered when applying ligand-based tools. Firstly, ligand alignments are based on the lowest conformation energy, which is often different from the bioactive conformation [244,245], as well as on the assumption that ligands bind in the same site and display the same conformation. Secondly, compounds should be evaluated by the same group (preferred) or tested using the same assay with the same parameters to be considered comparable [242]. Thirdly, the basic premise of ‘similar structures display similar activities’ are contradicted by the existence of activity cliffs [239,240,241], and so care should be taken when selecting potential candidates from a pool of virtual hits. Finally, it is also a challenge to incorporate the effects of solvation and protein flexibility due to the nature of the analysis.

As mentioned in the previous section, integration of several *in silico* methods have become common practice when designing and optimizing lead candidates to overcome the shortcomings of each individual tools. Despite requiring more computational resources, assimilation of computational methods result in better accuracy and enrichment of hits. In addition, the combination of a researcher’s innate knowledge with the computational efficiency of these tools is perhaps the best integration of all, as a human’s touch continues to be irreplaceable in the interpretation of all the data produced by *in silico* methods.

## 5. Conclusions and Future Perspectives

TB remains to be a relevant public health threat worldwide, necessitating accelerated discovery and design of novel antimycobacterial agents. Computer-aided drug design has become one of anchors of drug discovery research and continues to be a formidable tool in the hunt for promising drug leads, especially for tuberculosis. Continuous advancements in computing power and available software can enhance current computational tools and their application to different stages in the drug discovery pipeline. Nonetheless, these methods are not invincible as each tool have their own restrictions, and approximations are often used during the analysis. To overcome these, it is best to assimilate several *in silico* tools to complement the strength and limitations of each method used. The application of CADD in TB research has led to the identification of several antimycobacterial compounds that have already reached clinical evaluations, promoting its value in the drug discovery paradigm. Nonetheless, more work has to be done in order to expedite the discovery of anti-TB therapeutics.

Machine learning (ML) methods are making a comeback in drug discovery studies due to the upsurge in available data and enhanced computational powers. This has resulted in a wave drug discovery studies involving artificial intelligence (AI), wherein ML and deep learning (DL) techniques are applied to efficiently and ‘intelligently’ solve problems. This new shift in the drug discovery landscape is observed in personalized medicine and a number of relevant illnesses like cancer. While there are already several FDA-approved uses of AI in healthcare and diagnostics [301], it has yet to produce a successful drug candidate but it might not be far off. Currently, AI studies involving TB frequently covers diagnostics and treatment outcomes. This is perhaps one of the gaps that needs to be filled to be able to fast-track the discovery of novel and efficacious anti-TB drugs and finally alleviate the heavy burden of this infection around the globe.

## Figures and Tables

**Figure 1 molecules-25-00665-f001:**
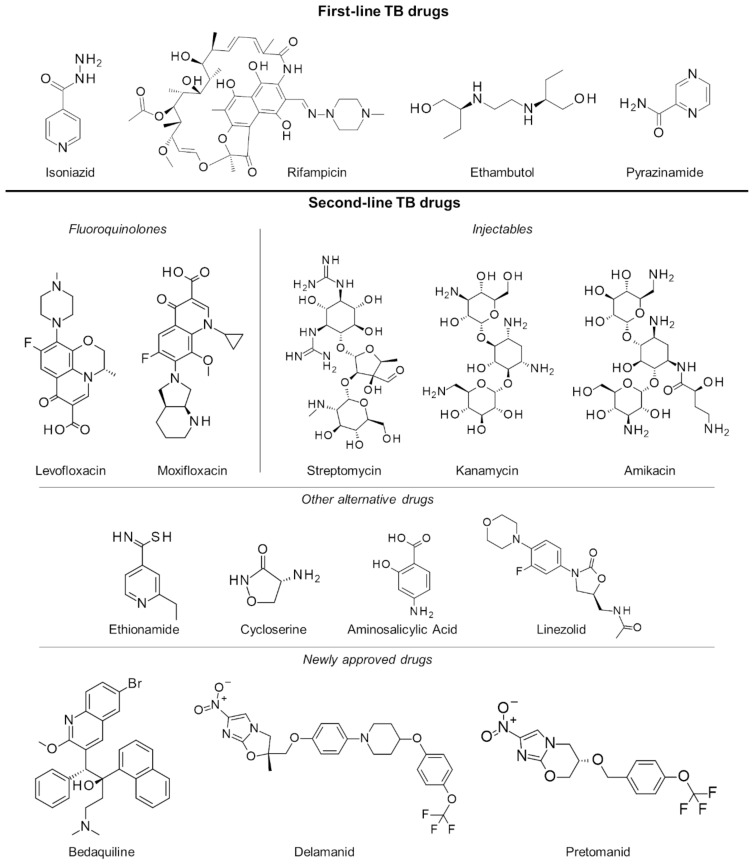
First- and second-line drugs approved for tuberculosis treatment.

**Figure 2 molecules-25-00665-f002:**
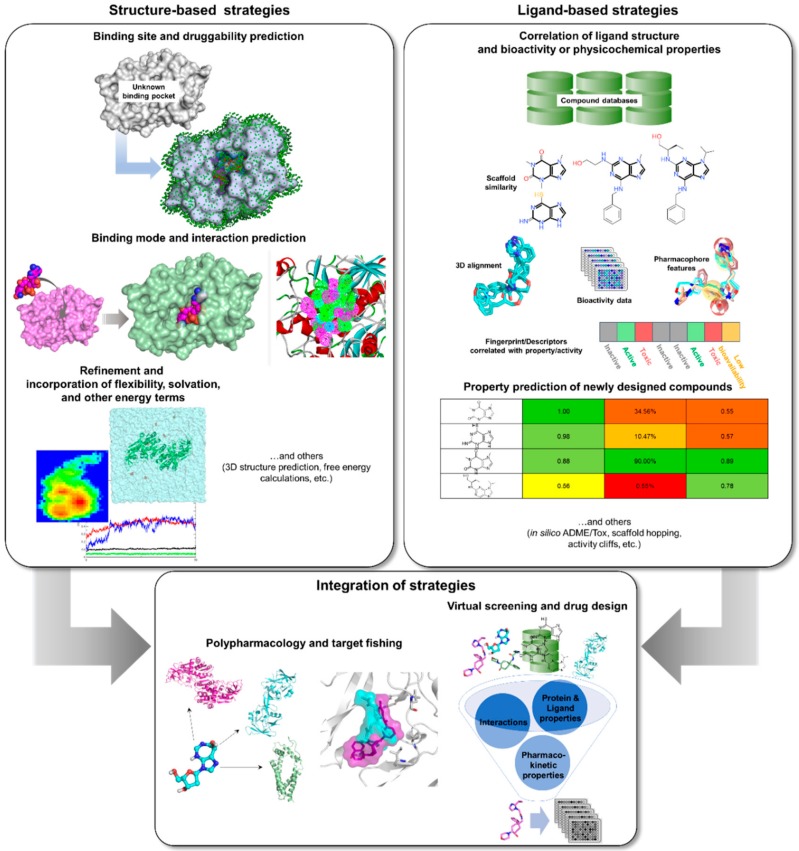
*In silico* tools that can be applied to TB drug design and development.

**Figure 3 molecules-25-00665-f003:**
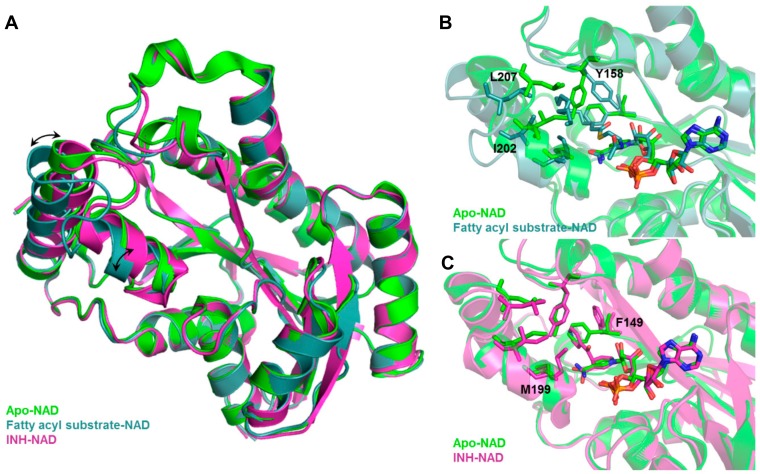
Backbone and sidechain flexibility shown by Mtb InhA, apo vs. fatty acyl-bound vs. INH-bound. (**A**) Structural overlay of apo, fatty acyl-bound, INH-bound Mtb InhA shows backbone movement upon substrate (fatty acyl) binding. Binding site comparison of (**B**) fatty acyl-bound and (**C**) INH-bound vs. apo Mtb InhA structure shows distinct changes in residue side chain positions and conformations. Black arrows indicate movement of alpha helices, side chains that showed large conformational change upon fatty acyl or INH binding are labelled in black.

**Figure 4 molecules-25-00665-f004:**
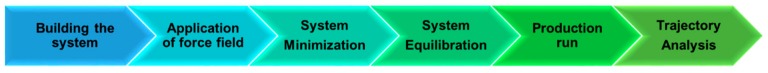
Typical molecular dynamics simulation workflow.

**Figure 5 molecules-25-00665-f005:**
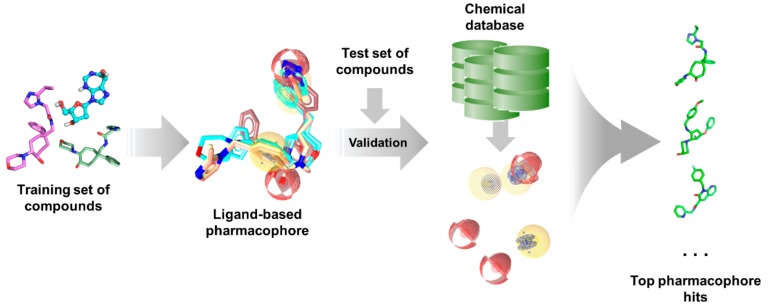
Typical ligand-based pharmacophore generation and screening workflow.

**Table 1 molecules-25-00665-t001:** Free and commercially available programs, webservers, and source codes for SBDD and LBDD.

Function	Software/ Webserver Name	Availability	Website
Comparative modeling	SWISS-MODEL [32]	Free webserver	https://swissmodel.expasy.org/
Structural geometry confirmation	MODELLER [33]	Free standalone program for academic license or commercially available through BIOVIA	https://salilab.org/modeller/ https://www.3dsbiovia.com/
	Robetta [34]	Free webserver	http://new.robetta.org/
	Prime [35]	Commercially available through Schrödinger	https://www.schrodinger.com/prime
	I-TASSER [36,37,38,39,40,41]	Free webserver or standalone program for academic license	https://zhanglab.ccmb.med.umich.edu/I-TASSER/
	HHPred [42,43,44]	Free webserver	https://toolkit.tuebingen.mpg.de/tools/hhpred
Structural geometry confirmation	PROCHECK [45]	Free webserver and source code	https://www.ebi.ac.uk/thornton-srv/software/PROCHECK/
Druggability and binding site prediction Druggability and binding site prediction	ProSA [46]	Free webserver	https://prosa.services.came.sbg.ac.at/prosa.php
	VERIFY3D [47]	Free webserver	https://servicesn.mbi.ucla.edu/Verify3D/
	ERRAT [48]	Free webserver	https://servicesn.mbi.ucla.edu/ERRAT/
	PockDrug [49]	Free webserver	http://pockdrug.rpbs.univ-paris-diderot.fr/cgi-bin/index.py?page=home
	DoGSiteScorer [50]	Free webserver	https://proteins.plus/
	fpocket [51,52]	Free/open source platform	https://github.com/Discngine/fpocket
	CASTp [53,54,55]	Free webserver	http://sts.bioe.uic.edu/castp/calculation.html
	PocketQuery [56]	Free webserver	http://pocketquery.csb.pitt.edu/
	PASS [57]	Free/open source platform	http://www.ccl.net/cca/software/UNIX/pass/overview.html
	SiteMap [58]	Commercially available through Schrödinger	https://www.schrodinger.com/sitemap
Docking, pharmacophore, and virtual screening Docking, pharmacophore, and virtual screening	ConCavity [59]	Free webserver	https://compbio.cs.princeton.edu/concavity/
	PrankWeb [60]	Free webserver	http://prankweb.cz/
	ProFunc [61]	Free webserver	http://www.ebi.ac.uk/thornton-srv/databases/ProFunc/
	AutoDock [62] and AutoDock Vina [63]	Free standalone program	http://autodock.scripps.edu/
	DOCK [64]	Free/open source platform	http://dock.compbio.ucsf.edu/
	GOLD [65]	Commercially available through CCDC	https://www.ccdc.cam.ac.uk/solutions/csd-discovery/components/gold/
	Glide [66]	Commercially available through Schrödinger	https://www.schrodinger.com/glide/
	Induced Fit [67]	Commercially available through Schrödinger	https://www.schrodinger.com/induced-fit
	FlexX [68]	Commercially available through BioSolveIT	https://www.biosolveit.de/flexx/index.html
	RosettaLigand [69]	Free/open source platform for academic license	https://www.rosettacommons.org/software
	CDOCKER [70]	Commercially available through BIOVIA	https://www.3dsbiovia.com/
	SwissDock [71,72]	Free webserver	http://www.swissdock.ch/docking
	Pharmer [73]	Free/open source platform	http://smoothdock.ccbb.pitt.edu/pharmer/
	CATALYST [74]	Commercially available through BIOVIA	https://www.3dsbiovia.com/products/collaborative-science/biovia-discovery-studio/pharmacophore-and-ligand-based-design.html
	PharmGist [75]	Free webserver	http://bioinfo3d.cs.tau.ac.il/pharma/php.php
	LigandScout [76]	Commercially available through Inte:Ligand	http://www.inteligand.com/ligandscout/
	SwissSimilarity [77]	Free webserver	http://www.swisssimilarity.ch/
	LEA3D [78]	Free webserver	https://chemoinfo.ipmc.cnrs.fr/LEA3D/index.html
	PyRx [79]	Free (no support) or commercially available	https://pyrx.sourceforge.io/
	Phase [80]	Commercially available through Schrödinger	https://www.schrodinger.com/phase
Molecular Dynamics	AMBER [81,82]	Commercially available	https://ambermd.org/
	CHARMM [83]	Free or commercially available through CHARMM or BIOVIA	http://charmm.chemistry.harvard.edu/ https://www.3dsbiovia.com/products/collaborative-science/biovia-discovery-studio/simulations.html
	CHARMMing [84]	Free webserver	https://www.charmming.org/charmming
	GROMACS [85,86]	Free/open source platform	http://www.gromacs.org/
	NAMD [87]	Free/open source platform	https://www.ks.uiuc.edu/Research/namd/
	Desmond [88]	Commercially available through Schrödinger	https://www.schrodinger.com/desmond
	SwissParam [89]	Free webserver	http://www.swissparam.ch/
	CHARMM-GUI [90]	Free webserver	http://www.charmm-gui.org/
	ParamChem CGenFF [91,92,93]	Free webserver	https://cgenff.umaryland.edu/
	VMD [94]	Free/open source platform	https://www.ks.uiuc.edu/Research/vmd/
Molecular Descriptors, Fingerprints, and Quantitative Structure-Activity Relationship	Dragon [95]	Commercially available through Talete	http://www.talete.mi.it/products/dragon_description.htm
	*E*-Dragon [96]	Free webserver	http://146.107.217.178/lab/edragon/start.html
	Canvas [97]	Commercially available through Schrödinger	https://www.schrodinger.com/canvas
	RDKit [98]	Free/open source platform	https://www.rdkit.org/docs/source/rdkit.ML.Descriptors.MoleculeDescriptors.html
	PyDescriptor [99]	Free/open source platform	https://ochem.eu/home/show.do
	Mordred [100]	Free/open source platform	https://github.com/mordred-descriptor/mordred
	Open3DQSAR [101]	Free/open source platform	http://open3dqsar.sourceforge.net/?Home
	ChemSAR [102]	Free webserver	http://chemsar.scbdd.com/
	SeeSAR [103]	Commercially available through BioSolveIT	https://www.biosolveit.de/SeeSAR/
Pharmacokinetic properties	QikProp [104]	Commercially available through Schrödinger	https://www.schrodinger.com/qikprop
	ADMET Predictor [105]	Commercially available through SimulationsPlus, Inc.	https://www.simulations-plus.com/software/overview/
	ACD Percepta [106]	Commercially available through ACD/Labs	https://www.acdlabs.com/products/percepta/index.php
	FAF-Drugs4 [107]	Free webserver	http://fafdrugs4.mti.univ-paris-diderot.fr/
	PatchSearch [108]	Free webserver	http://mobyle.rpbs.univ-paris-diderot.fr/cgi-bin/portal.py#forms::PatchSearch
	TOPKAT [109] and ADMET [110]	Commercially available through BIOVIA	https://www.3dsbiovia.com/products/collaborative-science/biovia-discovery-studio/qsar-admet-and-predictive-toxicology.html
	PASS Online [111]	Free webserver or commercially available standalone program	http://pharmaexpert.ru/Passonline/index.php
	SwissADME [112]	Free webserver	http://www.swissadme.ch/
	MetaSite [113]	Commercially available through Molecular Discovery	https://www.moldiscovery.com/software/metasite/
	ToxPredict [114]	Free webserver	https://apps.ideaconsult.net/ToxPredict#
	VirtualToxLab [115,116,117,118]	Free standalone software	http://www.biograf.ch/index.php?id=home
	admetSAR [119,120,121]	Free webserver	http://lmmd.ecust.edu.cn/admetsar1/home/
	MetaTox [122,123]	Free webserver	http://way2drug.com/mg2/

More available tools and detailed descriptions for the programs and servers can be found at https://www.click2drug.org/.

**Table 2 molecules-25-00665-t002:** Publicly available compound libraries.

Database	Size (Approximate)	Website
GDB-17 [140]	166 billion	http://gdb.unibe.ch/
Enamine REAL [141]	700 million	https://enamine.net/
PubChem [131]	97 million	https://pubchem.ncbi.nlm.nih.gov/
ChemSpider [142]	77 million	http://www.chemspider.com/
ZINC [143]	230 million	http://zinc.docking.org/
ChEMBL [144]	1.9 million	https://www.ebi.ac.uk/chembl/
NCI [145]	460,000	https://cactus.nci.nih.gov/download/roadmap/

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
