# Peer review of "In Silico Strategies in Tuberculosis Drug Discovery"

_molecules, 2020, doi:10.3390/molecules25030665_

Round 1

Reviewer 1 Report

The paper is well written and well organized. The review was very informative. 

The authors well reviewed tuberculosis and its drug resistance. Various computational strategies in TB drug discovery were reviewed with the proper details.

After minor revision including English, the paper should be accepted.

Some abbreviations need full descriptions such as TNF in line 48, VS in the line 255. 

Remove "calculations", in the line 384.

The authors may want to mention the speed up of MD using GPUs.

Author Response

Dear Reviewer,

Thank you for your valuable comments about our review article. As per your
suggestion, we have made changes in line with comments from the editors and
reviewers. Furthermore, we submitted our manuscript for similarity analysis and paraphrased several lines in the article (in red color font). You can view our point-by-point responses to your comments below (in blue color font). We hope that our revised manuscript satisfies the requirements of the editors and reviewers.

Thank you again for your time and consideration, and please let us know if you
need any further information.

Sincerely,

Stephani Joy Y. Macalino, Ph.D.

1. Some abbreviations need full descriptions such as TNF in line 48, VS in the line 255.

-> We have added the full description of TNF in line 48. VS was first fully described in line 144, but we realized that it is more proper to use the full term as the start of the paragraph in line 255 instead of the acronym. We have also checked other acronyms throughout the paper.

2. Remove "calculations", in the line 384.

-> We have removed this as per your suggestion.

3. The authors may want to mention the speed up of MD using GPUs.

-> We have included a brief description of MD acceleration using GPUs in lines 396-402.

Reviewer 2 Report

In my opinion, this review is well written and could be useful for the readers.

I have only one negative comment:

The authors did not report data about the development of salicylate synthase
MbtI inhibitors. This target is the first enzyme involved in the biosynthesis of mycobactins, compounds able to chelate iron, that is an essential cofactor for the survival of Mycobacterium tuberculosis in the host.

Author Response

Dear Reviewer,

Thank you for your valuable comments about our review article. As per your
suggestion, we have included a few more case studies in line with your comments. Furthermore, we submitted our manuscript for similarity analysis and paraphrased several lines in the article (in red color font). You can view our point-by-point responses (in blue color font) to your comments below. We hope that our revised manuscript satisfies the requirements of the editors and reviewers.

Thank you again for your time and consideration, and please let me know if you
need any further information.

Sincerely,

Stephani Joy Y. Macalino, Ph.D.

1. The authors did not report data about the development of salicylate synthase
MbtI inhibitors. This target is the first enzyme involved in the biosynthesis of mycobactins, compounds able to chelate iron, that is an essential cofactor for the survival of Mycobacterium tuberculosis in the host.

-> Thank you for your suggestion. A couple of case studies involving salicylate synthase MbtI inhibitor discovery were included in the review article describing latest inhibitors discovered for this target.

Round 2

Reviewer 2 Report

It can be accepted for publication